# Pemafibrate Protects Against Retinal Dysfunction in a Murine Model of Diabetic Retinopathy

**DOI:** 10.3390/ijms21176243

**Published:** 2020-08-28

**Authors:** Yohei Tomita, Deokho Lee, Yukihiro Miwa, Xiaoyan Jiang, Masayuki Ohta, Kazuo Tsubota, Toshihide Kurihara

**Affiliations:** 1Laboratory of Photobiology, Keio University School of Medicine, Tokyo 160-8582, Japan; y.tomita@keio.jp (Y.T.); deokholee@keio.jp (D.L.); yukihiro226@gmail.com (Y.M.); jxy@keio.jp (X.J.); 2Department of Ophthalmology, Keio University School of Medicine, Tokyo 160-8582, Japan; 3Boston Children’s Hospital/ Harvard Medical School, Boston, MA 02115, USA; 4Animal Eye Care, Tokyo Animal Eye Clinic, Tokyo 158-0093, Japan; 5Kowa Company, Ltd., Tokyo 103-8433, Japan; ms-ota@kowa.co.jp; 6Tsubota Laboratory, Inc., Tokyo 160-0016, Japan

**Keywords:** diabetes retinopathy (DR), electroretinography (ERG), fibroblast growth factor21 (FGF21), pemafibrate, streptozotocin (STZ), synaptophysin, selective peroxisome proliferator-activated receptor alpha modulator (SPPARMα)

## Abstract

Diabetic retinopathy (DR) is one of the leading causes of blindness globally. Retinal neuronal abnormalities occur in the early stage in DR. Therefore, maintaining retinal neuronal activity in DR may prevent vision loss. Previously, pemafibrate, a novel selective peroxisome proliferator-activated receptor alpha modulator, was suggested as a promising drug in hypertriglyceridemia. However, the role of pemafibrate remains obscure in DR. Therefore, we aimed to unravel systemic and retinal changes by pemafibrate in diabetes. Adult mice were intraperitoneally injected with streptozotocin (STZ) to induce diabetes. After STZ injection, diet supplemented with pemafibrate was given to STZ-induced diabetic mice for 12 weeks. During the experiment period, body weight and blood glucose levels were examined. Electroretinography was performed to check the retinal neural function. After sacrifice, the retina, liver, and blood samples were subjected to molecular analyses. We found pemafibrate mildly improved blood glucose level as well as lipid metabolism, boosted liver function, increased serum fibroblast growth factor21 level, restored retinal functional deficits, and increased retinal synaptophysin protein expression in STZ-induced diabetic mice. Our present data suggest a promising pemafibrate therapy for the prevention of early DR by improving systemic metabolism and protecting retinal function.

## 1. Introduction

Diabetic retinopathy (DR) is a common metabolic complication of diabetes, which has a significant impact on the world’s health systems [1]. DR often causes vision loss in people of working age, despite increasing evidence of the effectiveness of regular screening and early treatment for DR [1]. Most of the current treatments for DR try to regulate vascular changes through the vascular endothelial growth factor (VEGF), by laser treatment, and anti-VEGF drugs [2]. However, since anti-VEGF drugs can affect physiological roles of VEGF in the retina, it may affect the retinal and blood vessel homeostasis [2]. Thus, innovative and comprehensive approaches are needed to reduce the risk of vision loss, including degeneration of the neural retina as well as that of normal blood vessels. Dysfunction of retinal electrophysiology occurs in advance of vascular abnormalities in DR [3]. Therefore, boosting retinal function may help to prevent DR progression.

Recently, a novel selective peroxisome proliferator-activated receptor alpha (PPARα) modulator (SPPARMα) pemafibrate (mw: 490.55 g/mol, Parmodia®, K-877, Kowa, Tokyo, Japan) was approved in Japan as a therapeutic agent against hyperlipidemia. Pemafibrate reduces serum triglycerides and increases high-density lipoprotein cholesterol as a hypolipidemic agent by activating PPARα in the liver [4]. Several studies suggested pemafibrate improves lipid profiles and insulin resistance in patients with type 2 diabetes and hypertriglyceridemia [5]. Pemafibrate activates PPARα in human hepatocytes, resulting in an increase in fibroblast growth factor21 (FGF21) expression [6], and our group has shown that pemafibrate attenuates pathological neovascularization in the retina via increasing FGF21 level in the plasma and liver [7].

FGF21 is a protein that regulates important metabolic pathways [8]. FGF21 is expressed in various tissues, mainly in the liver, and circulates to interact with other sites to modulate metabolism [8]. FGF21 may be able to improve metabolic homeostasis in diabetes. FGF21 level in patients with type 1 diabetes is lower than that in healthy subjects [9,10], implying that it may be important to maintain FGF21 level for treatment of diabetes. FGF21 protects against renal dysfunction and testicular apoptotic cell death in murine models of type 1 diabetes, [11,12] and prevents diabetic nephropathy [13]. In the retina, long-acting FGF21 (PF-05231023) was reported to decrease pathological retinal vessel proliferation and promote physiological retinal vascularization in a murine oxygen-induced retinopathy model [14] and protect against retinal neural dysfunction in a type 1 diabetic mouse model [15]. Furthermore, our previous study showed that long-acting FGF21 prevented retinal vascular leakage induced by VEGF in vitro and in vivo [16]. The administration of FGF21 has therapeutic effects on lipid and glucose metabolism in mice. Furthermore, in type 2 diabetic patients, it improves the level of blood lipids [17]. However, long-term treatment of FGF21 and maintenance of a high level of FGF21 still have clinical limitations, which in turn may necessitate an alternative way to boost FGF21 level in the body, such as through administration of pemafibrate.

There is limited knowledge of the role of FGF21 and pemafibrate in DR. Therefore, in our current study, we aimed to unravel whether administration of pemafibrate induces plasma FGF21 level and has a protective effect on retinal function in a streptozotocin (STZ)-induced diabetic mouse model and clarify the mechanism underlying the effect.

## 2. Results

### 2.1. Pemafibrate Improves Diabetic Condition in an STZ-Induced Diabetic Mouse Model

To investigate the protective effects of pemafibrate against DR, we first established an STZ-induced diabetic mouse model (Figure 1A). On week 1 after the first STZ intraperitoneal injection, mice showed lower body weight and hyperglycemia (Figure 1B,C), which is commonly seen in STZ-induced diabetic mice [18]. Next, STZ-induced diabetic mice were randomly divided into two groups and fed a diet with pemafibrate (0.0002%) or vehicle. The dose of 0.0002% (1.67 mg/kG diet weight) was chosen, in that the value in serum concentration of pemafibrate in each mouse consistently met minimum detection criteria (Appendix A, Figure A1). During the experimental period, there was no significant difference in body weight between pemafibrate-administered and vehicle-administered STZ-induced diabetic mice (Figure 1B). On week 4, blood glucose level was significantly reduced by pemafibrate administration in STZ-induced diabetic mice (Figure 1C). Until 11 weeks, this tendency was mildly maintained, but the difference between the groups was not significant. 

### 2.2. Pemafibrate Suppresses Visual Dysfunction in an STZ-Induced Diabetic Mouse Model

To determine the protective effect of pemafibrate on visual function in STZ-induced diabetic retinas, we performed electroretinography (ERG) (Figure 2). Reduced oscillatory potential (OP) amplitude is known to occur in the early stage of murine and human DR [19,20,21,22]. Pemafibrate significantly reversed the reduction of OP1, OP2, OP3, and total OPs (ΣOPs) amplitudes in STZ-induced diabetic mice (Figure 2A). There was no remarkable difference in a-wave and b-wave amplitudes between pemafibrate-administered and vehicle-administered retinas (Figure 2B). Next, using the optical coherence tomography (OCT), the ganglion cell complex (GCC) was measured (Appendix B, Figure A2). The GCC thinning has been shown in STZ-induced diabetic mice [23] as well as patients with diabetes [24,25]. We also found that the GCC significantly decreased in the STZ-induced diabetic retinas (Appendix B, Figure A2). However, there was no change between the groups treated with vehicle and pemafibrate (Appendix B, Figure A2).

### 2.3. Pemafibrate Improves Lipid Metabolism in an STZ-Induced Diabetic Mouse Model 

It has been reported that PPARα decreases triglyceride levels and increases high-density lipoprotein cholesterol levels [4]. Therefore, we examined whether pemafibrate improves lipid metabolism in STZ-induced diabetic mice. Serum triglyceride levels and serum total cholesterol levels in pemafibrate-administered mice were significantly lower and higher than those in vehicle-administered mice, respectively (Figure 3A,B). Pemafibrate administration trended toward a decrease in β-Hydroxybutyrate level and non-esterified fatty acids (NEFA) concentration but the differences were not significant (Figure 3C,D).

### 2.4. Pemafibrate Boosts Liver Function and Promotes Gene Expression of PPARα Downstream in an STZ-Induced Diabetic Mouse Model

We explored the liver because pemafibrate is principally metabolized in the liver in STZ-induced diabetic mice. Pemafibrate significantly increased relative liver weight (Figure 4A), which is a well-known effect in response to PPARα stimulation in rodents [26,27]. Next, the gene expression of PPARα downstream was examined (Figure 4B). *Abca1* and *Vldlr* mRNA levels were significantly upregulated in pemafibrate-administered livers compared with those in vehicle-administered livers. There was an increasing tendency in *Fabp4* mRNA level, but the difference was not significant between the groups.

### 2.5. Pemafibrate Increases Serum FGF21 Level and Retinal Synaptophysin Expression in an STZ-Induced Diabetic Mouse Model 

We examined whether pemafibrate increases the FGF21 level in STZ-induced diabetic mice. Serum FGF21 level was significantly elevated in pemafibrate-administered mice, in comparison with that of vehicle-administered mice (Figure 5A). We explored the liver, known as a primary target organ of pemafibrate (Figure 5B). Even though there was no significant difference in *Fgf21* mRNA expression between pemafibrate- and vehicle-administered livers, *Fgf21* mRNA expression in pemafibrate-administered livers was relatively higher than that in vehicle-administered livers. We explored the retina to examine whether pemafibrate directly increases *Fgf21* mRNA expression in the diabetic retina (Figure 5C). In the retina, no significant increase in *Fgf21* mRNA expression was observed. 

Next, we investigated the molecular mechanism underlying pemafibrate-mediated reversal of retinal visual dysfunction in STZ-induced diabetic mice. We analyzed the regulation of the synaptic vesicle protein, synaptophysin. Synaptophysin is abundantly expressed in the inner retinal neurons, which are the cellular source for OPs [28,29,30]. Increased retinal expression of synaptophysin was observed in pemafibrate-administered retinas compared with that in vehicle-administered retinas (Figure 6A). To examine whether increased serum FGF21 level by pemafibrate administration directly acts on retinal cells, we used the in vitro culture system with pheochromocytoma (PC)12D neuronal cells (Figure 6B). Differentiated PC12D cells were treated with long-acting FGF21 (PF-05231023) for 24 h. In agreement with the in vivo data, long-acting FGF21 increased synaptophysin expression in the differentiated PC12D cells.

## 3. Discussion

We revealed that the administration of a selective PPARα modulator, pemafibrate, mildly improved blood glucose levels and protected retinal function in STZ-induced diabetic mice in the current study. In addition, significant increases in PPARα target gene expressions in the liver and elevation of serum FGF21 level were observed after systemic administration of pemafibrate. That is consistent with a previous report of pemafibrate increasing FGF21 expression in type 2 diabetes with hypertriglyceridemia [31].

In diabetic status, hyperglycemia is a key determinant of all complications, such as kidney diseases, eye diseases, and all types of neuropathy [32]. It causes metabolic oxidative stress throughout the whole body. Managing hyperglycemia itself can relieve metabolic stress from tissue injuries. Based on our current study, high blood glucose level under diabetic condition was reduced by the administration of pemafibrate only on week 4. After week 4, we were not able to see a significant difference in blood glucose levels between pemafibrate- and vehicle-administered STZ-induced diabetic mice. This reducing effect of pemafibrate on blood glucose level was seen in a diet-induced obesity mouse model [33] and nonalcoholic steatohepatitis mouse model [34]. Although PPARα activation was suggested to be involved in glucose homeostasis regulation [35,36,37], we were not able to see constant significant glucose level regulation by pemafibrate administration. There might be several reasons for this discrepancy such as differences in pemafibrate doses and in vivo disease models. Thus, pharmacological targeting of PPARα activation to reduce blood glucose level needs to be further studied.

As the hypolipidemic agent, pemafibrate lowers triglyceride concentration in the body, of which a high level is considered as a risk factor for the development of cardiovascular diseases [38] and increases high-density lipoprotein cholesterol levels [37]. In our study, we also reproduced this finding in STZ-induced diabetic mice: We found a decrease in triglyceride concentration and an increase in total cholesterol levels that may be inferred as an increase in high-density lipoprotein cholesterol levels by the administration of pemafibrate. The potential molecular mechanisms for the favorable effects of pemafibrate on triglyceride and lipoprotein metabolism could be explained with the increases in *Abca1* and *Vldlr* mRNA levels in the liver after the administration of pemafibrate, of which gene expressions are closely involved in the increase in high-density lipoprotein cholesterol levels and the reduction of triglyceride concentration, respectively [4]. Lipid abnormalities, such as high triglyceride and low high-density lipoprotein cholesterol levels, are typically attributed to insulin resistance, and they are the complications often noted in people with type 2 diabetes [39]. In patients with type 2 diabetes and hypertriglyceridemia, pemafibrate treatment ameliorated lipid abnormalities and was well tolerated over the long-term [40]. Therefore, pemafibrate could be a powerful drug as a hypolipidemic agent for balancing lipid metabolism in diabetes.

The retina is one of the highest oxygen and nutrition demanding tissues in the body [41]. As the energy demands for visual function are a substantial amount of oxygen and nutrition, high-energy consumption could make the retina highly vulnerable to metabolic oxidative stress by hyperglycemia in diabetes [42]. Dysfunction in retinal neurons is the early retinal change seen in diabetic patients [3]. Therefore, maintaining retinal neural function may prevent vision loss in DR. In our study, we found that the administration of pemafibrate reversed the diabetes-induced OP amplitude reduction. However, there was no rescue of the GCC area reduction. These data indicate that pemafibrate may rescue the diabetic retina functionally without affecting the morphological change. In addition, we found that pemafibrate increased retinal synaptophysin expression. Previous reports suggested the cellular origins of OPs are neurons with synapse formation in the inner retina including bipolar and amacrine cells [28,29,30], both of which develop a retinal neuronal network contributing to visual function, and synaptophysin is a well-known synaptic vesicle membrane protein important for a neuronal network. In the brain, synaptophysin expression was shown to decrease with the progressions of Alzheimer’s disease and Parkinson’s disease [43]. In the retina, pathogenic conditions such as retinal detachment [44] and retinal inflammation [45] caused the reduction of synaptophysin expression. In our previous paper, synaptophysin expression decreased in the retina under an STZ-induced diabetic condition [46]. Mice deficient in synaptophysin exhibited a significant decrease in the number of synaptic vesicles [47], which can cause visual dysfunction. Although it is possible for other cooperative involvement of several synaptic proteins to reverse OP amplitude reduction by pemafibrate administration in STZ-induced diabetic mice, at least, increase in retinal synaptophysin expression after systemic administration of pemafibrate could contribute to anti-retinal dysfunction.

A growing body of evidence suggests that FGF21 acts as a therapeutic factor under pathologic conditions in the central nervous system [48,49]. Previous studies have shown that FGF21 receptors were expressed in the brain [50,51]. It has also been confirmed that FGF21 could pass through the blood–brain barrier by simple diffusion [52]. FGF21 protects the blood–brain barrier via FGFR1/β-klotho, crucial receptors for FGF21 function, after traumatic brain injury [53], and acts on the suprachiasmatic nucleus in the hypothalamus and controls female reproduction, suggesting that FGF21 may be an important mediator in the liver–neuroendocrine axis [54]. In the retina, FGFR1 and β-klotho were also expressed [14], and FGF21 restored the retinal neuronal functional deficits detected by ERG and reduced an inflammatory marker *Il-1β* mRNA level in the diabetic retina [15]. In our current study, after the administration of pemafibrate, mRNA expression level of *Fgf21* was not increased in the retina. Instead, there was a tendency of an increase in *Fgf21* mRNA expression level in the liver and a significant increase in serum FGF21 level. This phenomenon was also observed in another mouse model of oxygen-induced retinopathy in our previous paper [7]. As the blood–retinal barrier, essential for normal visual function [55], is broken down in DR [56], the leaky blood–retinal barrier might allow circulating FGF21 to be transported easily from blood to the retina, and FGF21 may directly act on the retinal cells. Therefore, the increase in serum FGF21 level by pemafibrate administration could be a primary effector of boosting neuronal activity in the diabetic retina rather than an autocrine or a paracrine effector. However, we do not exclude the potential contribution of local FGF21. Moreover, vascular protective effects of FGF21 on the retinal endothelial cells [16], systemic metabolic changes by pemafibrate administration, and direct protective effects of pemafibrate on the retina [57] could also be taken into consideration, which needs to be further studied.

Emerging evidence suggests that PPARα agonists including fenofibrate may be useful for the treatment of DR [58]. However, in clinical uses, these drugs have been associated with an increased risk of kidney injury with increased levels of serum creatinine [59]. Fenofibrate is excreted through the kidneys, and the excretion decreases in patients with kidney dysfunction [59]. For this reason, patients with severe kidney dysfunction are recommended not to use fenofibrate and other fibrate PPARα drugs [60]. As a novel selective PPARα modulator, pemafibrate has higher potency and selectivity for PPARα activation than fenofibrate [6,61,62]. Pemafibrate was reported to exert more triglyceride concentration reducing and high-density lipoprotein cholesterol concentration increasing effects than that of fenofibrate [63]. Furthermore, pemafibrate showed fewer kidney-related adverse events than that of fenofibrate in clinical studies [64]. In terms of the protective effect for the retina, we previously reported that the administration of pemafibrate attenuated pathological neovascularization in the retina in a murine oxygen-induced retinopathy model, whereas the administration of fenofibrate could not [7]. However, direct comparison studies regarding diabetic retinal protection may need to be further studied for better clinical uses.

In conclusion, we developed an STZ-induced diabetic mouse model and provided those mice with diet supplemented with pemafibrate. We found that pemafibrate administration mildly reduced blood glucose level, improved lipid metabolism, boosted liver function, increased serum FGF21 level, and protected retinal neural activity (Figure 7). Even though more data are required for understanding the in vivo mode of action for FGF21 or other unknown effector molecules induced by pemafibrate, pemafibrate could be useful as a promising protective agent for DR.

## 4. Materials and Methods

### 4.1. Animal

Mice, 6 to 8 weeks old male C57BL/6, were purchased from CLEA Japan (Tokyo, Japan) and housed in a temperature-controlled environment with free access to food and water under a 12 h light–dark cycle. All animal experimental protocols were approved by the Ethics Committee on Animal Research of the Keio University School of Medicine (Approved number #16017-1). Procedures adhered to the ARVO Statement for the Use of Animals in Ophthalmic and Vision Research in accordance with the international standards of animal care and use, ARRIVE (Animal Research: Reporting in Vivo Experiments) guidelines (http://www.nc3rs.org.uk/arrive-guidelines).

### 4.2. Cell Line

The rat pheochromocytoma PC12D cell line was kindly supplied by Dr. Yoko Ozawa (Keio University, Japan) [65] and maintained in DMEM (Cat #08456-36, Nacalai Tesque, Kyoto, Japan) media supplemented with 10% FBS and 1% streptomycin-penicillin at 37 °C under an atmosphere containing 5% CO_2_. For subculture, PC12D cells were harvested at 80% confluence. For differentiation into neuronal cells, the cells were seeded at 1.0 × 10^6^ cells/well on 6-well tissue culture plates coated with Poly-L-lysine (Cat #P4707, Sigma, Tokyo, Japan). After 24 h, the cell culture medium was changed to DMEM containing 0.5% FBS and 50 ng/mL nerve growth factor (NGF, Cat #13257019, Thermo Fischer Scientific, Waltham, MA, USA). After 3 days, the cells were treated with 100 ng/mL long-acting FGF21 (PF-05231023, MedKoo Biosciences, Morrisville, NC, USA) in serum-free DMEM media for our experiments.

### 4.3. STZ-Induced DR Model and Administration of Pemafibrate 

STZ-induced DR was made according to the schedule depicted in Figure 1A. Mice were starved 3 h prior to intraperitoneal injection of 100 mg/Kg/day STZ. Diabetes was induced for 3 days of consecutive STZ injection. After 1 week, only mice with a blood glucose level over 300 mg/dl were chosen as the STZ-induced diabetic mice. STZ-induced diabetic mice were randomly divided into two groups: the vehicle-administered group and the pemafibrate-administered group. After grouping, mice were fed diet supplemented with vehicle (normal chow, CE-2; 5% fat, CLEA, Tokyo, Japan) or 0.0002% pemafibrate (pemafibrate, Kowa Co. Ltd. Tokyo, Japan; normal chow, CE-2; 5% fat, CLEA, Tokyo, Japan) until they were sacrificed. Body weight and blood glucose levels were measured during the experimental period and relative liver weight (liver weight/body weight) was measured after sacrifice.

### 4.4. Pemafibrate Detection

The serum concentration of pemafibrate was measured according to the manufacturer’s instruction (Kowa Co. Ltd. Tokyo, Japan). Briefly, 150 µL of blood samples were collected by orbital sinus puncture under pentobarbital anesthesia. Plasma samples were obtained by centrifugation (9000× *g*, 4 °C, 5 min). The concentration of pemafibrate in plasma was measured by liquid chromatography-tandem mass spectrometry after de-proteinization using methanol. Chromatographic separation of pemafibrate was achieved on a Kinetex C18 column (2.1 × 50 mm, 2.6 µm particle size; Phenomenex, Torrance, CA, USA) using 0.1% formic acid and acetonitrile in an Agilent 1100/1200 Series HPLC system (Agilent Technologies, Santa Clara, CA, USA).

### 4.5. Electroretinography (ERG)

ERG was performed as previously described [66]. Briefly, mice were dark adapted for at least 12 h, prepared under dim red illumination, and anesthetized with a combination of midazolam (Sandoz, Tokyo, Japan), medetomidine (Orion, Espoo, Finland), and butorphanol tartrate (Meiji Seika Pharma, Tokyo, Japan). They were placed on a heating pad throughout the experiment. Pupils were dilated with a drop of a mixture of 0.5% tropicamide and 0.5% phenylephrine (Santen Pharmaceutical, Osaka, Japan). Full field flash ERG responses were recorded using a Ganzfeld dome, an acquisition system, and LED stimulators (PuREC, MAYO, Inazawa, Japan). The active electrodes were recorded with contact lens electrodes and a reference electrode was placed subcutaneously between the eyes. A clipping electrode to the tail served as a ground. ERG responses were obtained from both eyes of each mouse. The amplitudes of a- and b-waves and OPs were measured and compared among age-matched non-diabetic mice (naive) and STZ-induced diabetic mice administered with vehicle or pemafibrate.

### 4.6. Metabolic Measurements

Serum levels of triglyceride, total cholesterol, β-hydroxybutyrate, NEFA, and FGF21 were measured according to the manufacturer’s instruction. Briefly, after blood sample collection, serum samples were enzymatically processed with Qualigent (triglyceride and total cholesterol, Sekisui Medical, Tokyo, Japan), 3HB-L Test Kit (β-hydroxybutyrate, Kainos, Tokyo, Japan), and Clinimate NEFA (NEFA, Sekisui Medical, Tokyo, Japan). Then, they were measured using the Hitachi Automated Analyzer (Labospect003, Hitachi High-Technologies, Tokyo, Japan). Serum FGF21 level was measured using an FGF-21 ELISA kit (Cat #RD291108200R, BioVendor Laboratory Medicine, Brno, Czech Republic).

### 4.7. Quantitative PCR

Quantitative PCR was performed as previously described [7]. Briefly, the total RNA of the retina was extracted using an RNeasy Plus Mini Kit (Qiagen, Velno, The Netherlands). RT-PCR was performed using a ReverTra Ace® qPCR RT Master Mix with gDNA Remover (TOYOBO, Osaka, Japan), and Real-time PCR was performed using a THUNDERBIRD® SYBR® qPCR Mix (TOYOBO, Osaka, Japan) with the Step One Plus Real-Time PCR system (Applied Biosystems, Waltham, MA, USA). Total RNA of the liver was extracted using ISOGEN II (Nippon Gene, Tokyo, Japan). RT-PCR was performed using High-capacity cDNA Reverse Transcription Kits (Life Technologies, Carlsbad, CA, USA). The Fast Real-Time PCR System (7900HT, Applied Biosystems, Waltham, MA, USA) was used for Real-time PCR using Fast SYBR® Green Master Mix (Applied Biosystems, Waltham, MA, USA). Primers used are listed in Table 1. The fold change between levels of different transcripts was calculated by the ΔΔC*_T_* method.

### 4.8. Western Blotting

Western blotting was performed as previously described [66]. Briefly, proteins were homogenized in lysis RIPA buffer (Cat #89900, Thermo Fischer Scientific, Waltham, MA, USA) containing a protease inhibitor cocktail (Cat #11836170001, Roche Diagnostics, Basel, Switzerland). After BCA assay for protein concentration, SDS loading buffer was added to protein lysates and the lysates were boiled at 95 °C for 3 min. The lysates were fractionated in 10% SDS-PAGE, transferred to PVDF membrane (PVDF, Merck, Darmstadt, Germany), and then blocked with 5% nonfat dry milk for 1 h. The membranes were incubated with primary antibodies, anti-Synaptophysin (1:1000, Cat #SAB4502906, Sigma, Tokyo, Japan), anti-α-Tubulin (1:1000, Cat #3873, Cell Signaling Technology, Danvers, MA, USA), or anti-β-Actin (1:1000, Cat #3700, Cell Signaling Technology, Danvers, MA, USA) at 4 °C overnight. Membranes were washed with TBST several times and incubated with HRP-conjugated secondary antibodies (1:5000, GE Healthcare, Chicago, IL, USA) for 2 h at room temperature. The signal was detected using an ECL kit (EzWestLumi plus, ATTO, Tokyo, Japan). The protein bands were visualized via chemiluminescence (ImageQuant LAS 4000 mini, GE Healthcare, Chicago, IL, USA). Blotting was quantified using NIH ImageJ software (National Institutes of Health, Bethesda, MD, USA).

### 4.9. Optical Coherence Tomography (OCT)

The inner retinal areas of the mice were measured with an SD-OCT system (Envisu R4310, Leica, Wetzlar, Germany). This measurement was performed under mydriasis, by 0.5% tropicamide and 0.5% phenylephrine (Santen Pharmaceutical, Osaka, Japan), and general anesthesia with a combination of midazolam (Sandoz, Tokyo, Japan), medetomidine (Orion, Espoo, Finland), and butorphanol tartrate (Meiji Seika Pharma, Tokyo, Japan). A cross section of the retina of 0.25 mm radius from the optic nerve was measured, and the area of the ganglion cell complex (GCC), which is from the nerve fiber layer (NFL) to the inner plexiform layer (IPL), was calculated using NIH ImageJ software (National Institutes of Health, Bethesda, MD, USA).

### 4.10. Statistical Analysis

Analyses of data from all the experiments were performed with GraphPad Prism 5 (GraphPad Software, San Diego, CA, USA). Statistical significance was calculated using Student’s *t*-test and one-way ANOVA followed by a Bonferroni post hoc test. The *p*-values of less than 0.05 were considered statistically significant.

## Figures and Tables

**Figure 1 ijms-21-06243-f001:**
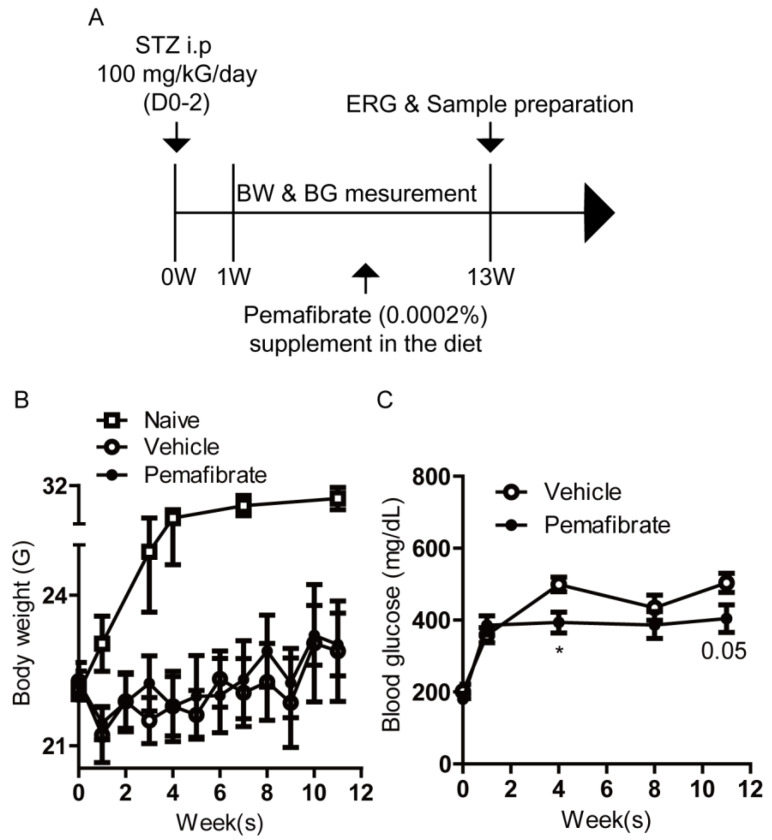
Schematic of the STZ-induced diabetic mouse model procedure and general observations. (**A**) A schematic illustration showing the STZ-induced diabetic model procedure and administration of pemafibrate to mice. BW; body weight, BG; blood glucose, ERG; electroretinography, STZ; streptozotocin, i.p; intraperitoneal injection. (**B**) Quantitative analysis (*n* = 3–8 per group) showed that body weight of the STZ-induced mice was lower than that of naive mice. There was no significant difference in body weight between pemafibrate-administered mice and vehicle-administered mice. (**C**) Quantitative analysis (*n* = 3–8 per group) showed that blood glucose level in pemafibrate-administered mice was lower than that in vehicle-administered mice only on week 4. * *p* < 0.05. Line graphs are presented as mean with ± standard deviation. The data were analyzed using Student’s *t*-test.

**Figure 2 ijms-21-06243-f002:**
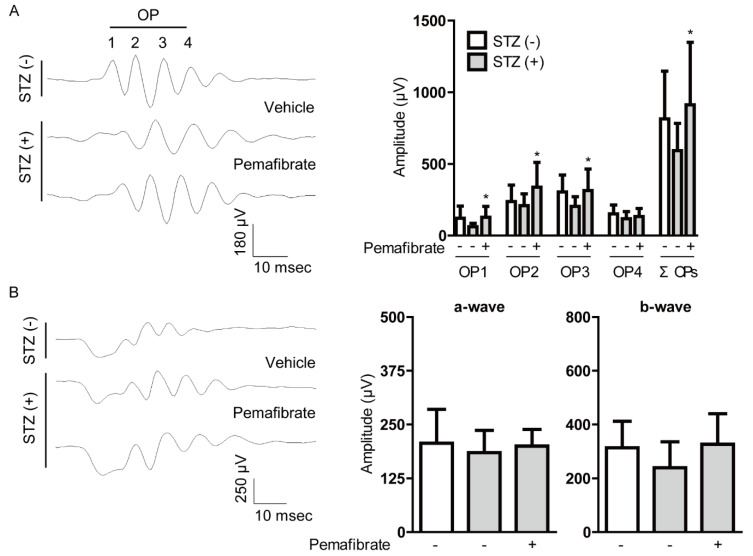
Change of visual function by pemafibrate in the STZ-induced diabetic mouse model. (**A**) Representative waveforms of OPs and quantitative analysis showed that pemafibrate reversed the reduction of OPs (OP1, OP2, OP3, and ΣOPs) amplitudes in STZ-induced diabetic mice (*n* = 10–16 per group). (**B**) Representative waveforms of a- and b-waves and quantitative analysis showed that pemafibrate did not increase the reduced amplitudes of a-wave and b-wave in STZ-induced diabetic mice (*n* = 14–18 per group). * *p* < 0.05, compared with vehicle-administered diabetic mice. Graphs are presented as mean with ± standard deviation. The data were analyzed using one-way ANOVA followed by a Bonferroni post hoc test. OP; oscillatory potential.

**Figure 3 ijms-21-06243-f003:**
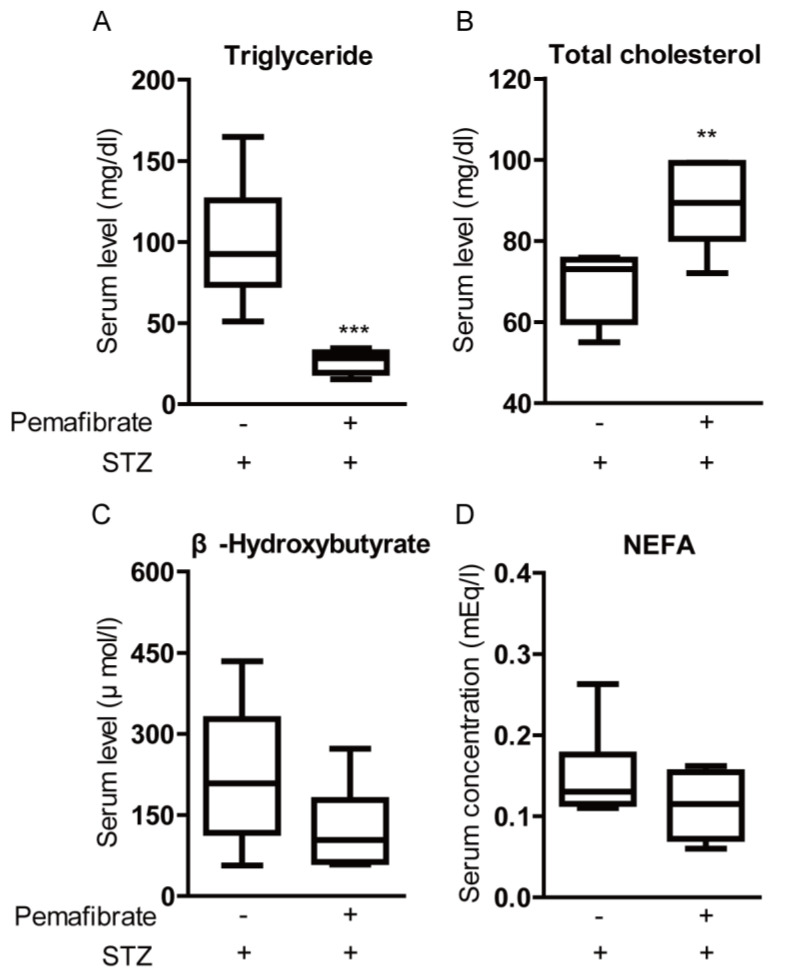
Change of lipid metabolism by pemafibrate in the STZ-induced diabetic mouse model. (**A**) Quantitative analysis showed that pemafibrate decreased serum triglyceride level in STZ-induced diabetic mice (*n* = 6 per group). (**B**) Quantitative analysis showed that pemafibrate increased serum total cholesterol level in STZ-induced diabetic mice (*n* = 6 per group). (**C**,**D**) There was no significant difference in serum level of β-Hydroxybutyrate and concentration of NEFA between pemafibrate-administered mice (*n* = 6) and vehicle-administered mice (*n* = 6). ** *p* < 0.01, *** *p* < 0.001. Graphs are presented as median with interquartile range, the 25th and 75th percentile. The data were analyzed using Student’s *t*-test. NEFA; non-esterified fatty acids.

**Figure 4 ijms-21-06243-f004:**
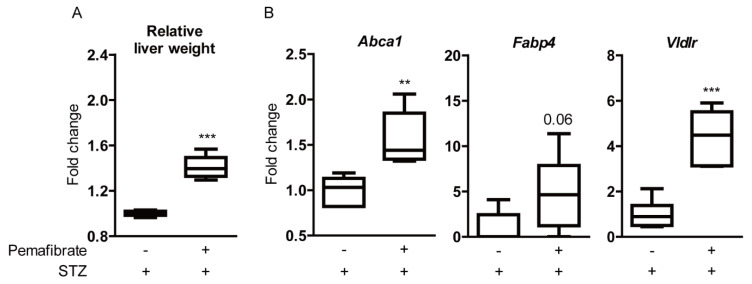
Increase of liver weight and PPARα downstream gene expression in the liver by pemafibrate in the STZ-induced diabetic mouse model. (**A**) Quantitative analysis (*n* = 6 per group) showed that relative liver weight (liver weight/body weight) in pemafibrate-administered mice was significantly higher than that in vehicle-administered mice. (**B**) Quantitative analysis showed that pemafibrate significantly increased *Abca1* and *Vldlr* mRNA levels and slightly increased *Fabp4* mRNA level in STZ-induced diabetic livers (*n* = 6 per group). ** *p* < 0.01, *** *p* < 0.001. Graphs are presented as median with interquartile range, the 25th and 75th percentile. The data were analyzed using Student’s *t*-test.

**Figure 5 ijms-21-06243-f005:**
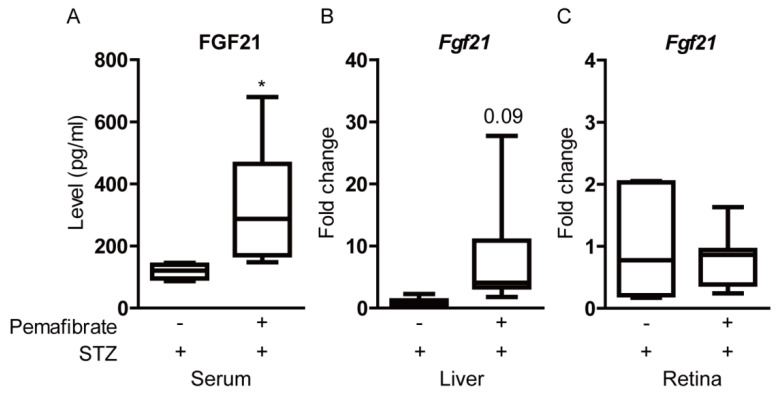
Increase of fibroblast growth factor21 (FGF21) level by the administration of pemafibrate in the STZ-induced diabetic mouse model. (**A**) Quantitative analysis showed that pemafibrate increased serum FGF21 level in STZ-induced diabetic mice (*n* = 5–6 per group). (**B**) Quantitative analysis showed that pemafibrate slightly increased *Fgf21* mRNA level in STZ-induced diabetic livers (*n* = 6–7 per group). (**C**) Quantitative analysis showed that pemafibrate did not increase *Fgf21* mRNA level in STZ-induced diabetic retinas (*n* = 9–12 per group). * *p* < 0.05. Graphs are presented as median with interquartile range, the 25th and 75th percentile. The data were analyzed using Student’s *t*-test.

**Figure 6 ijms-21-06243-f006:**
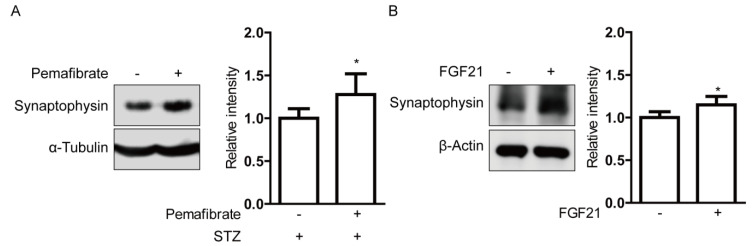
Increase in synaptophysin expressions by pemafibrate and fibroblast growth factor21 (FGF21) in STZ-induced diabetic retinas and differentiated pheochromocytoma (PC)12D cells. (**A**) Representative immunoblots and quantitative analysis for synaptophysin and α-Tubulin showed that pemafibrate significantly increased expression of synaptophysin in STZ-induced diabetic retinas (*n* = 4–7 per group). (**B**) Representative immunoblots and quantitative analysis for synaptophysin and β-Actin showed that long-acting FGF21 (100 ng/mL) significantly increased synaptophysin expression in differentiated PC12D cells (*n* = 5 per group). * *p* < 0.05. Bar graphs are presented as mean with ± standard deviation. The data were analyzed using Student’s *t*-test.

**Figure 7 ijms-21-06243-f007:**
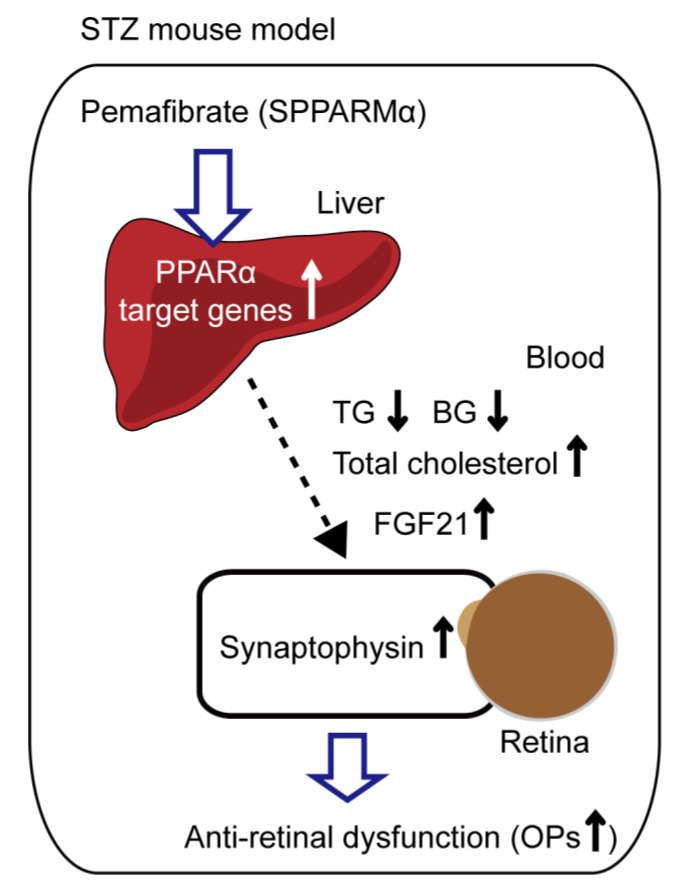
The working hypothesis of the protective mechanism against retinal dysfunction by the administration of pemafibrate in an STZ-induced diabetic mouse model. A possible mechanism for prevention against retinal dysfunction in diabetes is that systemic selective PPARα modulator (SPPARMα) pemafibrate administration boosts liver function and upregulates PPARα target genes in the liver, and an increased level of serum FGF21 improves lipid metabolism by decreasing triglyceride (TG) and increasing total cholesterol levels, mildly decreasing blood glucose level, and finally resulting in an increase in retinal synaptophysin expression and prevention against reduced amplitudes of OPs.

**Table 1 ijms-21-06243-t001:** Primer list.

Name	Direction	Sequence (5’ → 3’)	Accession Number
*β-actin*	Forward	GGGAAATCGTGCGTGACA	NM_007393.5
	Reverse	CAAGAAGGAAGGCTGGAAAA	
*Abca1*	Forward	CGTTTCCGGGAAGTGTCCTA	NM_013454.3
	Reverse	GCTAGAGATGACAAGGAGGATGGA	
*Fabp4*	Forward	CCGCAGACGACAGGA	NM_024406.3
	Reverse	CTCATGCCCTTTCATAAACT	
*Fgf21*	Forward	AACAGCCATTCACTTTGCCTGAGC	NM_020013.4
	Reverse	GGCAGCTGGAATTGTGTTCTGACT	
*Vldlr*	Forward	GAGCCCCTGAAGGAATGCC	NM_001161420.1
	Reverse	CCTATAACTAGGTCTTTGCAGATATGG	

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
