# Peer review of "Pemafibrate Protects Against Retinal Dysfunction in a Murine Model of Diabetic Retinopathy"

_ijms, 2020, doi:10.3390/ijms21176243_

Round 1

Reviewer 1 Report

The manuscript entitled “ Pemafibrate Protects Against Retinal Dysfunction in 2 a Murine Model of Diabetic Retinopathy” by Tomita el al. demonstrated interesting experimental data on diabetic retinopathy. Hypothesis, experiments, statistics, animal model, and data analysis are sound and accurate. It is worth of consideration for publication with contingency of providing that the authors will make the necessary revision as follows.

  1. Pemafibrate properties are not clear including concentration (mg/ml)? Molecular weight? Roughly how many molecules are there?
  2. Why vehicle-only reduces the body weight in Figure 1B?
  3. Any known anti-diabetic drug (Metformin, insulin, Sulfonylureas, Meglitinides, Thiazolidinediones, SGLT2 inhibitors, GLP-1 receptor agonists, DPP-4 inhibitors?) is recommended as a positive control to compare pemafibrate.
  4. What is the potential mechanism of increased cholesterol with pemafibrate in Figure 3B?
  5. It seems that the concentration of pemafibrate is too low. Dose- and time-dependent standard curves are strongly recommended for future studies.

There are minor revisions requested as follows with capital letters and underlines as suggestions.

26 performed to check up THE retinal neural function

After sacrifice, the retina, liver, (ADD COMMA) and blood samples

47 neural electrophysiological dysfunction occurs in THE advance of

55 resulting in increase in=INCREASING=resulting in AN increase in

71 long-term treatment of FGF21 and maintenance of A high level of FGF21

93 illustration showed THE STZ-induced

105 significantly reversed THE reduction of OP1, OP2, OP3

Figure 2. Change of visual function by pemafibrate in THE STZ-induced diabetic mouse model

120 decreasing triglyceride LEVELS and increasing high-density lipoprotein cholesterol LEVELS.

124 Pemafibrate administration trended toward A decrease in β-hydroxybutyrate level and non-esterified 125 fatty ACIDS (NEFA)

128 Figure 3. Change of lipid metabolism by pemafibrate in THE STZ-induced diabetic mouse model.

(C and D) There WAS no significant difference in serum level of

141 well-known effect in response to PPARα stimulation in rodents [23,24]. Next, THE gene expression of

158 We examined whether pemafibrate increases THE FGF21 level

179 Figure 5. Increase of fibroblast growth factor21 (FGF21) level by THE administration of pemafibrate in THE STZ-induced 180 diabetic mouse model.

195 0.05. Bar graphs were presented as mean with ± standard deviation. The data WERE analyzed using Student’s t-test.

209 Int. J. Mol. Sci. 2020, 21, x FOR PEER REVIEW 8 of 17 by THE administration of pemafibrate only on week 4. After week 4, we were not able to see A significant

222 high-density lipoprotein cholesterol levels, typically attribute

228 energy demands for visual function are A substantial amount of oxygen and nutrition

232 our study, we found that THE administration of

238 progressionS of

239 such as retinal detachment [41] and retinal inflammation [42] caused THE reduction

256 In our current study, after THE administration of

257 increased in the retina. Instead, there was a tendency of AN increase in

259 OTHER mouse model

269 In conclusion, we developed THE STZ-induced diabetic mouse model

278 Figure 7. THE working hypothesis of the protective mechanism against retinal dysfunction by THE administration

283 decreases blood glucose level, and finally results in AN increase in

289 hours light-dark cycle

315 Tokyo, Japan) until being sacrificed. Body weight and blood glucose LEVELS were

320 THE serum concentration of

322 puncture under pentobarbital anesthesia. Plasma samples were obtained by centrifugation (9,000×g, 323 4 ºC, 5 min). THE concentration of

344 Serum levels of triglyceride, total cholesterol, β-hydroxybutyrate, NEFA, (ADD COMMA) and FGF21 were

406 Figure A1. Detection of serum pemafibrate concentration in THE STZ-induced

Author Response

Response to Reviewer 1 Comments

Point 1:

The manuscript entitled “Pemafibrate Protects Against Retinal Dysfunction in a Murine Model of Diabetic Retinopathy” by Tomita el al. demonstrated interesting experimental data on diabetic retinopathy. Hypothesis, experiments, statistics, animal model, and data analysis are sound and accurate. It is worth of consideration for publication with contingency of providing that the authors will make the necessary revision as follows.

Pemafibrate properties are not clear including concentration (mg/ml)? Molecular weight? Roughly how many molecules are there?

Response 1:

Thank you for your valuable questions. The molecular weight of pemafibrate is 490.55 g/mol and a diet with pemafibrate (0.0002%) contains 1.67 mg of pemafibrate per 1 kg of a diet.

We added this description in the main text.

Introduction [line 49-50]

Recently, a novel selective peroxisome proliferator-activated receptor alpha (PPARα) modulator (SPPARMα) pemafibrate (mw: 490.55 g/mol, Parmodia®, K-877, Kowa, Tokyo, Japan)...

Results [line 83-84]

The dose of 0.0002% (1.67 mg/kg diet weight) was chosen in that the value in serum concentration of pemafibrate in each...

Point 2:

Why vehicle-only reduces the body weight in Figure 1B?

Response 2:

Thank you for your comments. The reduction or no change of body weight is a common result of the STZ-induced diabetic mouse model, which has been already known as one of the features after STZ injection as previously reported [1].

[1]          Furman, B.L. Streptozotocin-Induced Diabetic Models in Mice and Rats. Current protocols in pharmacology 2015, 70, 5.47.41-45.47.20, doi:10.1002/0471141755.ph0547s70.

This description was written in the manuscript.

Results_2.1. Pemafibrate Improves Diabetic Condition in STZ-Induced Diabetic Mouse Model [line 79-82]

To investigate the protective effects of pemafibrate against DR, we first established STZ-induced diabetic mouse model (Figure 1A). On week 1 after the first STZ intraperitoneal injection, mice showed lower body weight and hyperglycemia (Figure 1B, C), which is commonly seen in STZ-induced diabetic mice...

Point 3:

Any known anti-diabetic drug (Metformin, insulin, Sulfonylureas, Meglitinides, Thiazolidinediones, SGLT2 inhibitors, GLP-1 receptor agonists, DPP-4 inhibitors?) is recommended as a positive control to compare pemafibrate.

Response 3:

Thank you for your precious suggestion. We totally agree with your comment. However, we cannot complete those experiments at this point because of the limited revision period. We are going to follow up in our future study.

Point 4:

What is the potential mechanism of increased cholesterol with pemafibrate in Figure 3B?

Response 4:

Thank you for your valuable questions. The potential molecular mechanisms for the favorable effects of pemafibrate on triglyceride and lipoprotein metabolism could be explained with the increases in Abca1 and Vldlr mRNA levels in the liver after the administration of pemafibrate (Figure 4). Each Abca1 and Vldlr gene expression is closely involved in the increase in high-density lipoprotein cholesterol levels and the reduction of triglyceride concentration, respectively, as previously suggested and reported [1]. Therefore, we also think this is the potential mechanism why total cholesterol levels increased with pemafibrate treatment in our study.

[1]          Yamashita, S.; Masuda, D.; Matsuzawa, Y. Pemafibrate, a New Selective PPARα Modulator: Drug Concept and Its Clinical Applications for Dyslipidemia and Metabolic Diseases. Current Atherosclerosis Reports 2020, 22, 5, doi:10.1007/s11883-020-0823-5.

We added this description in the main text.

Discussion [line 220-229]

As the hypolipidemic agent, pemafibrate lowers triglyceride concentration in the body, of which high level is considered as a risk factor for the development of cardiovascular diseases [35], and increases high-density lipoprotein cholesterol levels [34]. In our study, we also reproduced this finding in STZ-induced diabetic mice: a decrease in triglyceride concentration and an increase in total cholesterol levels that may be inferred as an increase in high-density lipoprotein cholesterol levels by the administration of pemafibrate. The potential molecular mechanisms for the favorable effects of pemafibrate on triglyceride and lipoprotein metabolism could be explained with the increases in Abca1 and Vldlr mRNA levels in the liver after the administration of pemafibrate, of which gene expressions are closely involved in the increase in high-density lipoprotein cholesterol levels and the reduction of triglyceride concentration, respectively. Lipid abnormalities...

Point 5:

It seems that the concentration of pemafibrate is too low. Dose- and time-dependent standard curves are strongly recommended for future studies.

Response 5:

We thank the reviewer for the suggestion, and we understand the reviewer’s concern. Definitely, this suggestion will be reflected in our future studies.

Point 6:

There are minor revisions requested as follows with capital letters and underlines as suggesions.

26 performed to check up THE retinal neural function

After sacrifice, the retina, liver, (ADD COMMA) and blood samples

47 neural electrophysiological dysfunction occurs in THE advance of

55 resulting in increase in=INCREASING=resulting in AN increase in

71 long-term treatment of FGF21 and maintenance of A high level of FGF21

93 illustration showed THE STZ-induced

105 significantly reversed THE reduction of OP1, OP2, OP3

Figure 2. Change of visual function by pemafibrate in THE STZ-induced diabetic mouse model

120 decreasing triglyceride LEVELS and increasing high-density lipoprotein cholesterol LEVELS.

124 Pemafibrate administration trended toward A decrease in β-hydroxybutyrate level and non-esterified 125 fatty ACIDS (NEFA)

128 Figure 3. Change of lipid metabolism by pemafibrate in THE STZ-induced diabetic mouse model.

(C and D) There WAS no significant difference in serum level of

141 well-known effect in response to PPARα stimulation in rodents [23,24]. Next, THE gene expression of

158 We examined whether pemafibrate increases THE FGF21 level

179 Figure 5. Increase of fibroblast growth factor21 (FGF21) level by THE administration of pemafibrate in THE STZ-induced 180 diabetic mouse model.

195 0.05. Bar graphs were presented as mean with ± standard deviation. The data WERE analyzed using Student’s t-test.

209 Int. J. Mol. Sci. 2020, 21, x FOR PEER REVIEW 8 of 17 by THE administration of pemafibrate only on week 4. After week 4, we were not able to see A significant

222 high-density lipoprotein cholesterol levels, typically attribute

228 energy demands for visual function are A substantial amount of oxygen and nutrition

232 our study, we found that THE administration of

238 progressionS of

239 such as retinal detachment [41] and retinal inflammation [42] caused THE reduction

256 In our current study, after THE administration of

257 increased in the retina. Instead, there was a tendency of AN increase in

259 OTHER mouse model

269 In conclusion, we developed THE STZ-induced diabetic mouse model

278 Figure 7. THE working hypothesis of the protective mechanism against retinal dysfunction by THE administration

283 decreases blood glucose level, and finally results in AN increase in

289 hours light-dark cycle

315 Tokyo, Japan) until being sacrificed. Body weight and blood glucose LEVELS were

320 THE serum concentration of

322 puncture under pentobarbital anesthesia. Plasma samples were obtained by centrifugation (9,000×g, 323 4 ºC, 5 min). THE concentration of

344 Serum levels of triglyceride, total cholesterol, β-hydroxybutyrate, NEFA, (ADD COMMA) and FGF21 were

406 Figure A1. Detection of serum pemafibrate concentration in THE STZ-induced

Response 6:

Thank you very much for your comments. We revised all in our revised manuscript.

Reviewer 2 Report

The authors investigated pemafibrate could have the protective effect of retinal dysfunction in STZ-induced diabetic mice, and revealed that pemafibrate mildly improved blood glucose level and lipid metabolism, boosted liver function, increased serum FGF-21 level, increased retinal symaptophysin protein expression and restored retinal functional disorder due to STZ-induced diabetes. The current study was well-designed, and its results could demonstrate that pemafibrate could be a candidate drug for preventing from early diabetic retinal changes. The manuscript was also well-written, however, there were a few points to be addressed.

  1. The authors examined retinal functional changes using ERG, but there was no histopathological evaluation for retinal structural (i.e. neuronal and vascular) changes in STZ-induced diabetic mice. Were there any structural protective effect by pemafibrate? If yes, the authors should show the results and discuss the relationships between restore of retinal function and retinal structural changes.
  2. Some PPARα modulators such as fenofibrate have already had evidences for protective effect from diabetic retinal changes. What is the clinical strength of pemafibrate, which is a novel selective PPARα modulator approved newly in Japan? The authors should discuss the difference between the existing drugs and the new one especially in terms of the effect against the retina.

Author Response

Point 1:
The authors investigated pemafibrate could have the protective effect of retinal dysfunction in STZ-induced diabetic mice, and revealed that pemafibrate mildly improved blood glucose level and lipid metabolism,
boosted liver function, increased serum FGF-21 level, increased retinal symaptophysin protein expression and restored retinal functional disorder due to STZ-induced diabetes. The current study was well-designed, and its
results could demonstrate that pemafibrate could be a candidate drug for preventing from early diabetic retinal changes. The manuscript was also well-written, however, there were a few points to be addressed.
The authors examined retinal functional changes using ERG, but there was no histopathological evaluation for retinal structural (i.e. neuronal and vascular) changes in STZ-induced diabetic mice. Were there any structural
protective effect by pemafibrate? If yes, the authors should show the results and discuss the relationships between restore of retinal function and retinal structural changes.

Response 1:
Thank you very much for your valuable questions. We performed morphological evaluation utilizing OCT
(optical coherence tomography) to examine the inner retinal thickness change. Although the GCC (ganglion cell
complex) is reduced in the STZ-induced diabetic retina compared to non-diabetic controls, there is no change
between the groups treated with vehicle and pemafibrate as shown below. Taken together, the data indicated
that pemafibrate rescued the diabetic retina functionally without affecting the morphological change.

Image displayed in the attached file.

Point 2:
Some PPARα modulators such as fenofibrate have already had evidences for protective effect from diabetic retinal changes. What is the clinical strength of pemafibrate, which is a novel selective PPARα modulator
approved newly in Japan? The authors should discuss the difference between the existing drugs and the new one especially in terms of the effect against the retina.

Response 2:
Thank you for your important comments. As you mentioned above, some PPARα modulators such as fenofibrate have had evidence for a protective effect from diabetic retinal changes. However, clinically, fenofibrate has been associated with an increased risk of kidney injury with increased levels of serum creatinine [1]. fenofibrate is excreted through the kidneys and its excretion decreases in patients with compromised kidney function [1]. In this reason, patients with severe kidney dysfunction could be suggested not to use fenofibrate and other fibrate PPARα drugs [2].
As a novel selective PPARα modulator, pemafibrate has higher potency and selectivity for PPARα activation than fenofibrate [3-5]. Pemafibrate was suggested to exert more triglyceride concentration reducing and highdensity lipoprotein concentration increasing effects more than fenofibrtae [6]. Furthermore, pemafibrate showed less kidney-related adverse events than fenofibrate in clinical studies [7].
In terms of the protective effect against the retina, we previously reported that the administration of pemafibrate attenuated pathological neovascularization in the retina in a murine oxygen-induced retinopathy model whereas fenofibrate could not [8].

Taken together, we think that these factors are the clinical strength of pemafibrate.
[1] Davidson, M.H.; Armani, A.; McKenney, J.M.; Jacobson, T.A. Safety Considerations with Fibrate
Therapy. The American Journal of Cardiology 2007, 99, S3-S18, doi:doi.org/10.1016/j.amjcard.2006.11.016.
[2] Emami, F.; Hariri, A.; Matinfar, M.; Nematbakhsh, M. Fenofibrate-induced renal dysfunction, yes or no?
J Res Med Sci 2020, 25, 39-39, doi:10.4103/jrms.JRMS_772_19.
[3] Fruchart, J.-C. Pemafibrate (K-877), a novel selective peroxisome proliferator-activated receptor alpha
modulator for management of atherogenic dyslipidaemia. Cardiovascular Diabetology 2017, 16, 124,
doi:10.1186/s12933-017-0602-y.
[4] Raza-Iqbal, S.; Tanaka, T.; Anai, M.; Inagaki, T.; Matsumura, Y.; Ikeda, K.; Taguchi, A.; Gonzalez, F.J.;
Sakai, J.; Kodama, T. Transcriptome Analysis of K-877 (a Novel Selective PPARα Modulator (SPPARMα))-
Regulated Genes in Primary Human Hepatocytes and the Mouse Liver. Journal of atherosclerosis and
thrombosis 2015, 22, 754-772, doi:10.5551/jat.28720.
[5] Fruchart, J.C. Selective peroxisome proliferator-activated receptor α modulators (SPPARMα): the next
generation of peroxisome proliferator-activated receptor α-agonists. Cardiovasc Diabetol 2013, 12, 82,
doi:10.1186/1475-2840-12-82.
[6] Yamazaki, Y.; Abe, K.; Toma, T.; Nishikawa, M.; Ozawa, H.; Okuda, A.; Araki, T.; Oda, S.; Inoue, K.;
Shibuya, K., et al. Design and synthesis of highly potent and selective human peroxisome proliferator-activated
receptor alpha agonists. Bioorganic & medicinal chemistry letters 2007, 17, 4689-4693,
doi:10.1016/j.bmcl.2007.05.066.
[7] Ishibashi, S.; Arai, H.; Yokote, K.; Araki, E.; Suganami, H.; Yamashita, S. Efficacy and safety of
pemafibrate (K-877), a selective peroxisome proliferator-activated receptor α modulator, in patients with
dyslipidemia: Results from a 24-week, randomized, double blind, active-controlled, phase 3 trial. Journal of
clinical lipidology 2018, 12, 173-184, doi:10.1016/j.jacl.2017.10.006.
[8] Tomita Y, Ozawa N, Miwa Y, et al. (2019) Pemafibrate Prevents Retinal Pathological
Neovascularization by Increasing FGF21 Level in a Murine Oxygen-Induced Retinopathy Model. Int J Mol Sci
20(23). 10.3390/ijms20235878
We added thee descriptions in the manuscript.
Discussion [line 277-290]

Emerging evidence suggests that PPARα agonists including fenofibrate may be useful for the treatment of DR
[55]. However, in clinical uses, these drugs have been associated with an increased risk of the kidney injury
with increased levels of serum creatinine [56]. Fenofibrate is excreted through the kidneys and the excretion
decreases in patient with kidney dysfunction [56]. In this reason, patients with severe kidney dysfunction could
be suggested not to use fenofibrate and other fibrate PPARα drugs [57]. As a novel selective PPARα modulator,
pemafibrate has higher potency and selectivity for PPARα activation than fenofibrate [58-60]. Pemafibrate was
reported to exert more triglyceride concentration reducing and high-density lipoprotein cholesterol
concentration increasing effects than fenofibrtae [61]. Furthermore, pemafibrate showed less kidney-related
adverse events than fenofibrate in clinical studies [62]. In terms of the protective effect against retina, we
previously reported that the administration of pemafibrate attenuated pathological neovascularization in the
retina in a murine oxygen-induced retinopathy model whereas the administration of fenofibrate could not [7].
However, direct comparison studies regarding the diabetic retinal protection may need to be further studied
for better clinical uses.

Reviewer 3 Report

The authors of the article demonstrated that administration of a selective PPARα modulator, pemafibrate, mildly improved blood glucose level and protected retinal function in streptozotocin-induced
diabetic mice. During the experiment period, body weight and blood glucose level were examined. After sacrifice, the retina, liver and blood samples were subjected to the analyses and the authors found that pemafibrate mildly improved blood glucose level as well as lipid metabolism, boosted liver function, increased serum fibroblast growth factor21 level, restored retinal functional deficits and increased retinal synaptophysin protein expression in STZ-induced diabetic mice. Also the working hypothesis of the protective mechanism against retinal dysfunction was proposed. 

In addition, a few comments for the article, which needs to be improved:

  1. Please, characterize in more detail changes in lipid methabolism, for example, by full lipidomic analysis.
  2. Please, add western-blot experimental results for fig. 4 and 5.  

Author Response

Response to Reviewer 3 Comments

Point 1:

The authors of the article demonstrated that administration of a selective PPARα modulator, pemafibrate, mildly improved blood glucose level and protected retinal function in streptozotocin-induced

diabetic mice. During the experiment period, body weight and blood glucose level were examined. After sacrifice, the retina, liver and blood samples were subjected to the analyses and the authors found that pemafibrate mildly improved blood glucose level as well as lipid metabolism, boosted liver function, increased serum fibroblast growth factor21 level, restored retinal functional deficits and increased retinal synaptophysin protein expression in STZ-induced diabetic mice. Also the working hypothesis of the protective mechanism against retinal dysfunction was proposed.

In addition, a few comments for the article, which needs to be improved:

Please, characterize in more detail changes in lipid methabolism, for example, by full lipidomic analysis.

Please, add western-blot experimental results for fig. 4 and 5.

Response 1:

Thank you for your valuable comments. We totally agree with your comments which can further support our hypothesis. However, we cannot conduct these experiments because of the limited revision period given by the editor. We are going to follow up in our future studies. We appreciate your helpful suggestions.